# Efficacy of Colchicine in the Treatment of COVID-19 Patients: A Systematic Review and Meta-Analysis

**DOI:** 10.3390/jcm11092615

**Published:** 2022-05-06

**Authors:** Carlos J. Toro-Huamanchumo, Jerry K. Benites-Meza, Carlos S. Mamani-García, Diego Bustamante-Paytan, Abraham Edgar Gracia-Ramos, Cristian Diaz-Vélez, Joshuan J. Barboza

**Affiliations:** 1Escuela de Medicina, Universidad Cesar Vallejo, Trujillo 13007, Peru; toro2993@hotmail.com; 2Sociedad Científica de Estudiantes de Medicina de la Universidad Nacional de Trujillo, Trujillo 13011, Peru; jbenitesm@unitru.edu.pe; 3Facultad de Medicina Humana, Universidad Nacional de San Agustín, Arequipa 04002, Peru; cmamaniga@gmail.com; 4Facultad de Medicina Humana, Universidad de San Martín de Porres, Lima 15024, Peru; diegoandreebp@gmail.com; 5National Medical Center “La Raza”, Department of Internal Medicine, General Hospital, Instituto Mexicano del Seguro Social, Mexico City 06720, Mexico; dr.gracia.dmm@gmail.com; 6Universidad Privada Antenor Orrego, Trujillo 13007, Peru; cristiandiazv@hotmail.com; 7Instituto de Evaluación de Tecnologías en Salud e Investigación, Essalud, Lima 14072, Peru; 8Universidad Señor de Sipan, Chiclayo 14000, Peru; 9Tau-Relaped Group, Trujillo 13007, Peru

**Keywords:** colchicine, COVID-19, SARS-CoV-2, systematic review, meta-analysis

## Abstract

Objective: We assessed the efficacy of colchicine in COVID-19 patients through a systematic review. Methods: Six databases were searched until March 2022 for studies assessing colchicine versus control in hospitalized patients with COVID-19. The primary outcome was mortality, and secondary outcome was length of hospitalization. Inverse variance and random effect meta-analyses were performed. The strength of evidence was assessed using GRADE. Results: Nine studies (five randomized clinical trials (RCTs) and four non-randomized studies of intervention (NRSI); *n* = 13,478). Colchicine did not reduce mortality in comparison with the standard of care in RCTs (RR 0.99; 95%CI 0.90 to 1.10; *p* = 0.90); however, it did reduce mortality in NRSI studies (RR 0.45; 95%CI 0.26 to 0.77; *p* = 0.02). In the analysis of RCTs, colchicine did not reduce the length of hospitalization in comparison with the standard of care (MD: −2.25 days; 95%CI: −9.34 to 4.84; *p* = 0.15). Most studies were scored as having a high risk of bias. Quality of evidence was very low for primary and secondary outcomes. Conclusion: Colchicine did not reduce the mortality and length of hospitalization in comparison with the standard of care in hospitalized patients with COVID-19. The published evidence is insufficient and of very low quality to recommend treatment in patients with COVID-19.

## 1. Introduction

The worldwide pandemic caused by COVID-19 keeps the scientific-medical community uncertain due to the lack of a specific treatment protocol because there is no cure for the coronavirus [1]. In our country, Peru, treatment with colchicine has not been established for outpatients or hospitalized patients with COVID-19. Thus, in the last year, experimental vaccines and treatments have been studied and developed to combat SARS-CoV-2, as well as the characteristics of the viral infection [2]. Similarly, among the treatments evaluated in published clinical trials, immunomodulatory drugs against inflammatory reaction and cytokine storm in patients with severe and critical infection stand out. [3].

Colchicine, an immunomodulatory drug that acts by inhibiting microtubules, is widely used in conditions such as gout and those involving local tissue inflammation [4]. The rationale for using colchicine in patients with COVID-19 is based on the activation of the NLRP3 inflammasome by viroporin E, a component of SARS-CoV-2 that generates an inflammatory response [5]. Since colchicine inhibits the NLRP3 inflammasome, it has been postulated for use in SARS-CoV-2 infection [6].

The objective of this study was to assess the efficacy of colchicine in COVID-19 patients through a systematic review and meta-analysis.

## 2. Materials and Methods

This systematic review was reported following a PRISMA 2020 statement [7]. We assessed studies that evaluated the efficacy of treatment with colchicine in patients with COVID-19. The protocol was registered in the PROSPERO database (CRD42021230362).

### 2.1. Data Sources

We searched in PubMed, Scopus, Web of Science, Ovid-Medline, Embase, and Cochrane Central Register of Controlled Trials (CENTRAL) until 10 March 2022. The search strategy for PubMed was adapted for use in the other databases (Appendix A). There were no restrictions on language or publication date. We hand-searched reference lists of all included studies and relevant review articles to identify other potentially eligible trials. Additionally, we searched in the trial registries ClinicalTrials.gov (https://www.clinicaltrials.gov/, accessed on 10 March 2022), WHO International Clinical Trials Registry Platform (http://apps.who.int/trialsearch/, accessed on 10 March 2022), and a preprints/preproofs repository for finished as well as ongoing trials.

### 2.2. Eligibility Criteria

Studies were included if they met the following criteria: (i) randomized clinical trials (RCTs) and observational studies (non-randomized studies of intervention (NRSI)); (ii) hospitalized adult patients (≥18 years of age) diagnosed with COVID-19 as defined according to study authors; (iii) any dose and duration of colchicine as experimental/intervention group; (iv) placebo or standard of care as the control group or comparator. We excluded studies according to the following criteria: systematic reviews, narrative reviews, conference proceedings, editorials, case reports, case series, letters to the editor and abstracts.

### 2.3. Study Selection

One author (J.J.B.) downloaded all titles and abstracts retrieved by electronic searching to the Rayyan platform and duplicate records were removed. Titles and abstracts were independently screened for relevance by three review authors (J.B.M., D.B.P. and C.S.M.G.) and any disagreements were resolved by a fourth review author (J.J.B.). We retrieved the full text of selected trials and three authors (J.B.M., D.B.P. and C.S.M.G.) independently screened the full text, identified studies for inclusion, and registered reasons for the exclusion of studies. We resolved any disagreement through consulting a fourth review author (J.J.B.).

### 2.4. Outcomes

The primary outcome was mortality, and the secondary outcome was length of hospitalization.

### 2.5. Data Extraction

Three review authors (J.B.M., D.B.P. and C.M.) independently carried out data extraction using a data extraction form that was previously piloted on at least one study in the review and any disagreements were resolved by a fourth review author (J.J.B.). If additional data were needed, we contacted the corresponding author through email to request further information. We extracted the following study data from full-text articles: first author name, year of publication, study design, study location, study design, eligibility criteria, sample size, age, sex, description of intervention and control groups, primary and secondary outcomes.

### 2.6. Risk of Bias Assessment

Two investigators (D.B.P., C.S.M.G.) independently assessed risk of bias (RoB) by using the Risk of Bias in Non-Randomized Studies of Interventions (ROBINS-I) tool for NRSI [8] and the Cochrane Risk of Bias 2.0 tool for RCTs [9]; disagreements were resolved by discussion with a third investigator (J.J.B.). RoB per domain and study was described as low, moderate, serious, critical and no information for cohort studies, and as low, some concerns, and high for RCTs.

### 2.7. Statistical Analysis

Inverse variance and random effect meta-analyses were performed to evaluate the effect of colchicine vs. control on outcomes when outcome data were available for at least two RCTs or NRSI judged to have homogeneous study characteristics. Effects of meta-analyses were reported as relative risks (RR) and their 95% confidence intervals (CIs). CIs of effects were adjusted with the Hartung–Knapp method [10], and the between study variance tau2 was calculated with the Paule–Mandel method [11]. The effects of colchicine were described with log relative risks (LogRRs) with 95% confidence intervals (Log RR 95% CIs) for dichotomous outcomes in the NRSI studies that were evaluated. The RR (TE) and standard error (seTE) were calculated for the effect value in each study. Heterogeneity of effects among studies was quantified with the I^2^ statistic (an I^2^ > 60% means high heterogeneity). In sensitivity analyses, we assessed (i) all meta-analyses performed without the Hartung–Knapp adjustment and (ii) only studies with a low risk of bias. The R 3.5.1 meta-package was used for all meta-analyses. Statistical significance was set with a *p*-value < 0.05.

### 2.8. GRADE Quality of Evidence

The quality/certainty of evidence was evaluated using the GRADE methodology, which covers 5 aspects: risk of bias, inconsistency, indirectness, imprecision, and publication bias [12]. Quality of evidence was evaluated per outcome and described in Summary of Findings (SoF) tables; GRADEpro GDT was used to create SoF tables [13].

### 2.9. Ethical Considerations

This is a systematic review of published and open information in which no human subjects participated. Thus, no ethics committee approval was required.

## 3. Results

### 3.1. Selection of Studies

A total of 799 articles were identified in six databases; 493 duplicates articles were removed. Of 306 screened abstracts, 291 were excluded. Thus, 15 full-text studies were assessed for eligibility and 6 were excluded. Finally, nine studies (five RCTs and four NRSI; *n* = 13,478) were included for qualitative and quantitative analyses [14,15,16,17,18,19,20,21,22] (Figure 1).

### 3.2. Characteristics of Included Trials

Studies were conducted in USA [14,15], Italy [16,17], Greece [18], Brazil [19], UK–Indonesia–Nepal [20], México [21], and Argentina [22]. The mean age was 63 years (SD: 6.4). Three RCTs [18,19,20] and four NRSI [14,15,16,17] were included. Patients hospitalized with moderate to severe COVID-19 were included in all studies. The reported follow-up time in NRSI was between 14 and 28 days, and in RCTs it was between 21 and 28 days. Regarding the confounding analysis methods, propensity score matching adjusted by variables [14,17,20], Cox regression [16], and non-adjusted analysis [15,16,19,21,22] were applied (Table 1). Start doses of colchicine was 1.5 mg, 1.2 mg, 1 mg/day, 0.6 mg, and 0.5 mg. Other treatments such as hydroxychloroquine, azithromycin, remdesivir, and tocilizumab were used. Our search for ongoing trials identified 17 registered RCTs evaluating the effect of colchicine in hospitalized COVID-19 patients (Appendix A).

### 3.3. Risk of Bias Assessment

Overall, two RCTs were scored as high RoB [18,19] and one was scored as some concerns RoB [20]. One study had a high RoB in deviations from the intended interventions [18], and one study had a high RoB in missing outcome data (Appendix A). In NRSIs, two studies were scored as serious ROBINS-I [14,15] and one was scored as critical ROBINS-I [17].

### 3.4. Effect of Colchicine in Primary Outcomes

In the analysis of randomized controlled trials, colchicine did not reduce the mortality in comparison with the standard of care (RR 0.99; 95%CI 0.90 to 1.10; *p* = 0.90; I^2^ = 0%, Figure 2a). However, in the analysis of NRSI studies, colchicine reduced mortality in comparison with the standard of care (RR: 0.45; 95%CI: 0.26 to 0.77; *p* = 0.02; I^2^ = 22.6%; Figure 2b).

### 3.5. Effect of Colchicine in Secondary Outcomes

In the analysis of RCTs, colchicine did not reduce the length of hospitalization in comparison with the standard of care (MD: −2.25 days; 95%CI: −9.34 to 4.84; *p* = 0.15; I^2^ = 0%; Figure 3). For the other prespecified secondary outcomes in the protocol, there was insufficient information among the included studies to analyze the effects of colchicine on clinical improvement, the need for mechanical ventilation, transfer to the intensive care unit, serum levels of inflammatory markers, (C-reactive protein (CRP), D-dimer (DD), ferritin and lactate dehydrogenase (LDH)), serum levels of cardiac markers (troponin), and adverse effects.

### 3.6. Sensitivity Analysis

Sensitivity analysis showed no differences with the primary analysis for the outcomes evaluated (Appendix A).

### 3.7. Quality of Evidence (QoE)

QoE was very low for primary and secondary outcomes (Appendix A). In mortality, for RCTs and NRSI studies, the QoE was very low due to high risk of bias, the heterogeneity among the studies, and the imprecision of the effect. In length of hospitalization, the QoE was very low due to the high risk of bias and the imprecision of the effect.

## 4. Discussion

### 4.1. Main Findings

In our systematic review, we found that, in RCTs, colchicine did not reduce the mortality and length of hospitalization in comparison with the standard of care in hospitalized patients with moderate to severe COVID-19. However, in NRSI, colchicine reduced mortality in comparison with the standard of care. Most studies were scored as having a high risk of bias and the quality of evidence was very low for all outcomes.

Colchicine is a microtubule inhibitor that has been proposed as a possible treatment for COVID-19 patients based on the following mechanisms: (a) changes in SARS-CoV-2 viral replication due to changes in microtubules, which are important for intracellular essential transport and the creation of double-membrane vesicles [5]; (b) inhibition of the NLRP3 inflammasome with a decrease in interleukin (IL)-1β and the consequent reduction of several pro-inflammatory cytokines that are produced in excess in COVID-19 patients [23]; (c) reduced expression of L-selectin inhibits neutrophil activation, motility, and activation to cell endothelium [24]; (d) inhibition of neutrophil extracellular traps (NET) generation; (e) avoid more leukocyte migration and microvascular thrombosis due to inhibition of complement activation via microtubule disruption, inhibition of C5a, and reduced expression of C5aR; (f) endothelial damage is prevented indirectly by reducing E-selectin-mediated neutrophil adhesion to pro-inflammatory, cytokine-activated endothelial cells and limiting excessive inflammatory activation [25].

Many observational and experimental studies have been conducted based on this to examine the efficacy and safety of colchicine in a variety of COVID-19 scenarios. To examine the evidence from these, systematic reviews and meta-analysis are required. However, owing to the differences in technique, the results of numerous systematic reviews differ from ours.

Salah et al. published a meta-analysis about colchicine in COVID-19 patients and their primary outcomes were all-cause mortality and mechanical ventilation. The study included eight studies (three RCT, five observational, *n* = 5259). Colchicine showed a reduction in all-cause mortality (RR: 0.62; 95%CI: 0.48–0.8), but it did not reduce the risk of mechanical ventilation (RR: 0.75; 95%CI: 0.45–1.25) [26]. This study included both designs of studies (RCT and observational) in the meta-analysis, therefore the analysis had some concerns. In practice, it is not feasible to include data from different study designs in the same meta-analysis, because this may generate bias or results that are far from the true effect. In our study, the mortality was analyzed by type of study, and we report two effect measures. Another observation is related to the intervention. Our study included only colchicine as intervention; however, in Salah et al.’s work, the authors included studies that analyzed colchicine plus other treatments as an intervention. This may cause bias in the primary analysis of the outcome.

Moreover, Golpour et al. carried out a meta-analysis that included 10 studies (of which four were RCT, *n* = 5901) to evaluate the efficacy of colchicine in COVID-19 patients. Colchicine was associated with a decreased mortality rate in COVID-19 patients (RR: 0.365; 95%CI: 0.555–0.748) and a decrease in hospitalization time in COVID-19 patients [27]. Both Salah et al. and Golpour et al. did not report the quality of evidence. It is possible that, in both studies, the measure of the effect of “mortality” has been estimated or reported erroneously, therefore it is not possible to have adequate certainty to recommend treatment. Regarding the mortality in both studies, the RCTs included in these studies did not have a significant effect. As with our study, colchicine did not reduce the mortality compared with control based on RCTs effect measures.

Other studies have similar biases in their analysis. For example, Elshafei et al. evaluated nine studies (three RCT, six observational studies, *n* = 5522) and found lower mortality with colchicine use (OR: 0.35; 95%CI: 0.25–0.48). The qualities of the most included studies were rated as moderate by the authors [28]. Beran et al. analyzed eight studies (three RCT, five observational studies, *n* = 926) and showed a lower risk of mortality with the colchicine treatment (RR:0.49; 95%CI: 0.34–0.72). The included studies were rated as high quality by the authors [29]. In all previous studies, the authors combined both randomized and non-randomized studies in their meta-analyses. The inclusion of non-randomized studies should be limited to particular situations because the potential biases are greater for non-randomized studies compared with randomized trials when evaluating the effects of interventions [30].

Nawangsih et al. published a systematic review with a meta-analysis to evaluate the effect of colchicine administration on mortality in patients with COVID-19, which included eight studies (three RCT, five observational studies, *n* = 5530). As with our study, the authors carried out the analysis by type of study. The pooled analysis for observational studies showed mortality reduction (OR: 0.48; 95%CI: 0.28–0.82). However, pooled RCTs did not show this reduction (OR: 0.43; 95%CI: 0.17–1.08) [31]. Unlike the Nawangsih et al. study, our study did not include the COLCORONA trial (*n* = 4159), as we did not include outpatient studies. Nevertheless, our study included the RECOVERY trial (*n* = 11,340). This difference in sample size allows our study to have a more precise estimate of the effect of the intervention.

The Pan American Health Organization (PAHO) published a living systematic review and meta-analyses to evaluate the evidence of potential therapeutics options for COVID-19, including colchicine [32]. In this regard, they analyzed five RCT (*n* = 16,105), including the COLCORONA trial and RECOVERY trial [33]. As with our result in the pooled analysis of RCT, they found that colchicine did not reduce mortality (RR: 1; 95%CI: 0.93–1.08; moderate certainty). In addition, the living systematic review by the PAHO showed that colchicine does not reduce mechanical ventilation requirements (RR: 1.02; 95%CI: 0.95–1.13; moderate certain), though probably reduces hospitalizations in patients with the recently onset disease (RR: 0.8; 95%CI: 0.62–1.03), although the certainly of the evidence was low for this outcome because of imprecision. A living systematic review allows for the updating of the evidence regularly, which is important in the current pandemic scenario.

In our meta-analysis, differences were found between the incidence of sex-matched outcomes. However, other reports, both systematic reviews and other study designs, have not further analyzed the gender-matched analysis, therefore they have not noted a great importance in these differences in treatment with colchicine.

Our study has several strengths. First, we performed a recent and broad systematic search in six databases, two trials registers, and one preprints/preproofs repository without language restriction. Second, we evaluated RCTs and NRSI studies separately; the combination of all types of designs (as occurred in several previous studies) can increase the bias and confusion of the findings. Third, we evaluated the risk of bias from included studies and informed the results together with the certainty of the evidence using the GRADE methodology; meeting the methodological guidance for a high-quality systematic review and meta-analysis is mandatory to increase confidence in the findings. Fourth, we only evaluated studies with hospitalized patients to analyze a population of patients as homogeneous as possible; our findings do not support the use of colchicine in moderate to severe COVID-19. Finally, we also performed sensitivity analysis; the effect was the same as the primary analysis on mortality.

### 4.2. Limitations

Our study also has some limitations. The quality of evidence was very low for the outcomes; the quality of evidence limits the confidence that the estimates of the effect are correct. However, we evaluated the best currently available evidence about colchicine use in hospitalized patients with COVID-19. The result of a pooled analysis from NRSI studies differed from that obtained in the pooled RCTs; this discrepancy may be explained by a higher risk of bias in the NRSI studies that may overestimate the result [34]. Some outcomes were scarce in evaluating their impact in patients treated with colchicine. Well-designed, placebo-controlled clinical trials are needed to increase the certainty of the evidence about the effect of colchicine in COVID-19 patients.

## 5. Conclusions

Colchicine did not reduce the mortality and length of hospitalization in comparison with the standard of care in hospitalized patients with COVID-19. The published evidence is insufficient and of very low quality to recommend treatment in patients with COVID-19.

## Figures and Tables

**Figure 1 jcm-11-02615-f001:**
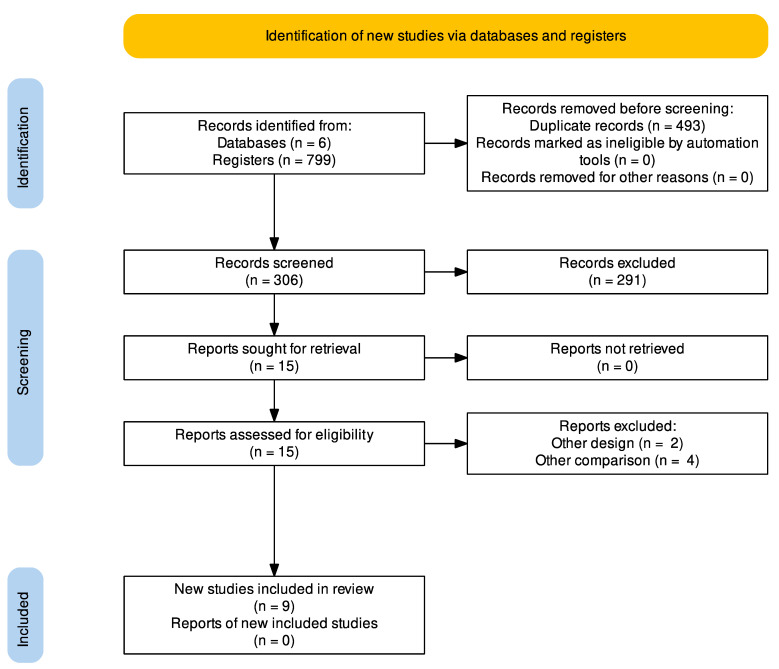
PRISMA flow chart of the studies selection process.

**Figure 2 jcm-11-02615-f002:**
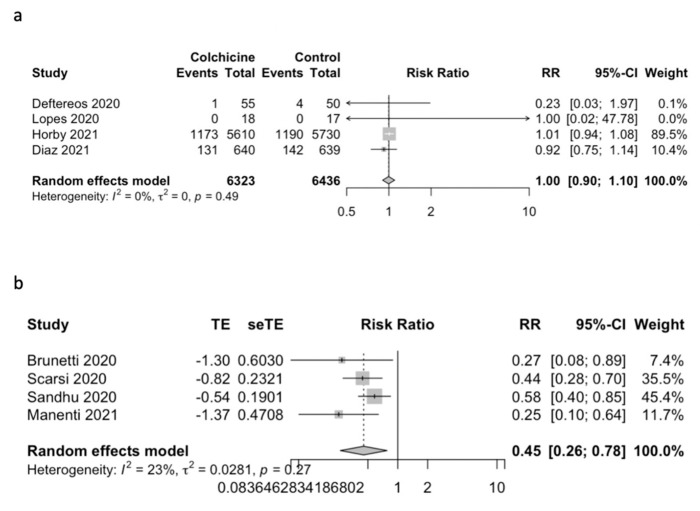
Forest plot of primary outcome: (**a**) RCT studies; (**b**) NRSI studies.

**Figure 3 jcm-11-02615-f003:**
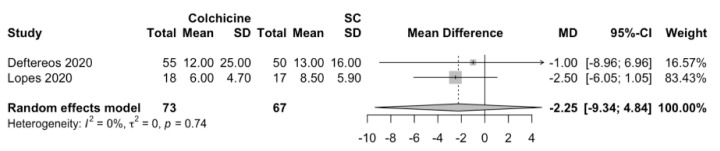
Forest plot of secondary outcome: length of hospitalization (in days).

**Table 1 jcm-11-02615-t001:** Characteristic of included studies.

Author, Year	Country	Design	Number of Patients	Type of Patients	Gender per Arm (Male, *n*,%)	Confounding Analysis Methods	Intervention	Comparator/Control	Other Treatments	Mortality, per Arm (*n*,%)	Length of Hospitalization, per Arm (Mean, SD)	Conclusions	Strength of Evidence
Brunetti, 2020 [14]	USA	NRSI	66	Severe COVID-19 patients with confirmed SARS-CoV-2 infection (positive PCR)	E: 21 (63.6) C: 22 (66.7)	Propensity score matching adjusted by age, sex, body mass index (BMI), select baseline laboratory values, baseline oxygen saturation on room air, receipt of tocilizumab, receipt of remdesivir, and comorbidity score.	Colchicine: Loading dose of 1.2 mg. Maintenance dose was 0.6 mg twice daily. Treatment was initiated within 72 h of hospital admission.	Standard care	Hydroxychloroquine, Azytromycin, Remdesivir, Tocilizumab	E: 3/33 (9.1) C: 11/33 (33.3)	NR	Colchicine is associated with lower mortality compared to standard treatment (OR 0.20, 95%CI 0.05–0.80, *p* = 0.023).	Low
Scarsi, 2020 [16]	Italy	NRSI	162	Hospitalised patients affected by COVID-19	E: 77 (63.0) C: 90 (64.0)	Cox proportional hazards regression survival analysis adjusted by demographical, clinical and laboratory parameters, comorbidities, and other treatments.	Colchicine 1 mg/day (reduced to 0.5 mg/day, if severe diarrhea).	Standard care	Antiviral drugs; Hydroxychloroquine; Corticosteroids	E: 20/122 (16.4) C: 52/140 (37.1)	E: 21.3 (6.8) C: 25 (14.8)	Colchicine is associated with improve outcomes in patients with COVID-19	Low
Sandhu, 2020 [15]	USA	NRSI	197	Patients clinically suspected COVID-19, or a positive SARS-CoV-2 nasal swab PCR	E: 21 (61.8) C: 40 (51.3)	No cofounding-adjusted analysis was applied.	Colchicine 0.6 mg twice a day for three days and then 0.6 mg once a day for a total of 12 days.	Standard care	Hydroxychloroquine, Steroids, Insulin, Oseltamivir, Enoxaparin, Direct acting oral anticoagulants, Intravenous heparin, Subcutaneous heparin, Warfarin	E: 16/34 (47.1) C: 63/78 (80.8)	E: 10.11 (median) C: 11 (median)	Colchicine improved outcomes in patients with COVID-19 receiving standard of care therapy	Low
Manenti, 2021 [17]	Italy	NRSI	141	Adult inpatients with a diagnosis of COVID-19 pneumonia based on: (1) CT typical findings, (2) positive nasopharyngeal swab test, and/or (3) serologic anti-SARS-CoV-2 antibody test.	E: 51 (72.9) C: 49 (69.0)	Propensity score matching adjusted by age, sex, categorical variate indicating the severity of conditions at onset namely, non-hospitalized, hospitalized without oxygen, hospitalized, and requiring supplemental oxygen, hospitalized requiring noninvasive ventilation, shortness of breath, cough, history of diabetes, history of hypertension, history of cancer, use of antibiotics, use of anti-retroviral drugs, use of hydroxychloroquine, use of i.v. steroids, use of tocilizumab.	Orally 1 mg/day from day 1 up until clinical improvement or up to a maximum of 21 days, according to physicians’ preferences. Doses were adjusted for kidney function and drug to drug interaction. The dose had to be reduced to 0.5 mg/day if the patient developed severe diarrhea.	Standard care	Antibiotics, Antiviral treatment, Hydroxychloroquine, IV steroids, Tocilizumab	E: 5/70 (7.1) C: 20/71 (28.2)	NR	Colchicine administration was associated with a significant reduction in mortality and accelerated clinical improvement, compared with control group. Also, colchicine reduced levels of CRP, lymphocyte count and IL-6. Colchicine has a well-known safe toxicity profile.	Moderate
Deftereos, 2020 [18]	Greece	RCT	105	Hospitalized adult patients diagnosed with SARS-CoV-2 infection, confirmed with polymerase chain reaction–reverse transcriptase testing.	E: 31(56.4) C: 30(60)	No cofounding-adjusted analysis was applied.	Colchicine administration (1.5 mg loading dose followed by 0.5 mg after 60 min) and maintenance doses of 0.5 mg twice daily) with standard medical treatment for as long as 3 weeks.	Standard care	Chloroquine or Hydroxychloroquine, Azytromycin, Lopinavir or ritonavir, Tocilizumab, Concomitant anticoagulation	E: 1/55 (1.8) C: 4/50 (8)	E: 12 (25) C: 13 (16)	Participants who received colchicine had statistically significantly improved time to clinical deterioration.	Low
Lopes, 2020 [19]	Brazil	RCT	72	Hospitalized with moderate or severe forms of COVID-19 diagnosed by RT-PCR in nasopharyngeal swab specimens and lung computed tomography scan involvement compatible with COVID-19 pneumonia	E: 9 (52.9) C: 5 (27.8)	No cofounding-adjusted analysis was applied.	Colchicine 0.5 mg thrice daily for 5 days, then 0.5 mg twice daily for 5 days	Placebo and standard of care	Azithromycin, hydroxychloroquine, heparin, Methylprednisolone, supplemental oxygen	E: 0/18 (0) C: 0/17 (0)	E: 6 (4.7) C: 8.5 (5.9)	The use of colchicine reduced the length of both, supplemental oxygen therapy and hospitalization.	Low
Díaz, 2021 [22]	Argentina	RCT	1279	Hospitalized adults (age >18 years) with confirmed or suspected SARS-CoV-2 infection were eligible for the trial if they were admitted to the hospital with symptoms suggestive of COVID-19	E: 421 (65.8) C: 409 (64.0)	No cofounding-adjusted analysis was applied.	Colchicine 1.5 mg, followed by 0.5 mg orally within 2 h of the initial dose, and subsequently 0.5 mg orally twice a day for 14 days or discharge	Usual care	NR	E: 131/640 C: 142/639	NR	Colchicine did not significantly reduce mechanical ventilation or 28-day mortality in patients hospitalized with COVID-19 pneumonia.	Low
Absalon, 2021 [21]	México	RCT	116	Hospitalized adult patients aged 18 to 70 years who tested positive for at least one of the following COVID-19	E: 37 (66) C: 39 (65)	Cox proportional hazards regression model and calculated hazard ratios (HR) with 95% CI.	Colchicine 1.5 mg, followed 0.5 mg PO BID for 10 days	Placebo	NR	NR	NR	Colchicine is safe but not effective in the treatment of severe COVID-19.	Low
Horby, 2021 (RECOVERY trial) [20]	United Kingdom, Indonesia, and Nepal	RCT	11340	Hospitalized patients with clinically suspected or laboratory confirmed SARS-CoV-2 infection and no medical history that might, in the opinion of the attending clinician, put the patient at significant risk if they were to participate in the trial.	E: 3896 (69.0) C: 4012 (70.0)	Statistical test of interaction, adjusted by age, sex, ethnicity, level of respiratory support, days since symptom onset, and use of corticosteroids.	Colchicine 1 mg after randomization followed by 500 mcg 12 hours later and then 500 mcg twice daily by mouth or nasogastric tube for 10 days in total or until discharge, whichever occurred earlier.	Standard of care	Corticosteroids	E: 1173/5610 (21.0) C: 1190/5730 (21.0)	E: 10 (median) C: 10 (median)	Colchicine was not associated with reductions in mortality, duration of hospitalization or the risk of being ventilated or dying for those not on ventilation at baseline. The results do not support the use of colchicine in adults hospitalised with COVID-19 and there is no clinical benefit compared with current usual care.	Low

## Data Availability

Not applicable.

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
