# Peer review of "Efficacy of Colchicine in the Treatment of COVID-19 Patients: A Systematic Review and Meta-Analysis"

_jcm, 2022, doi:10.3390/jcm11092615_

Round 1

Reviewer 1 Report

The authors studied the efficacy of colchicine in the treatment of hospitalized Covid-19 patients by conducting meta-analysis of both RCT and observational studies. The primary outcome was mortality and the secondary outcome only included hospitalization days. This review followed the PRISMA guideline and was registered in the PRSPERO database. The RCT and non-RCT studies were analyzed separately for the primary outcome and GRADE methodology was also used to improve the quality of this systematic review. The authors concluded that colchicine did not reduce the primary and secondary outcomes compared with the usual care.

This study was properly conducted and provided a conclusion different from those of other reviews the authors cited. There was not enough good evidence suggesting the therapeutic effects of colchicine and more RCTs are needed.

Although the authors included hospitalized patients only to make the study population as homogeneous as possible, important data were omitted as a result. Many patients were unable to be hospitalized at the peak of this Covid-19 crisis. 

Deftereos 2020 and Horby 2021 provided data on the need for mechanical ventilation and should be studied and included in this review since it is an important outcome.

Author Response

Dear reviewer. 

Regarding your comment: 

"Deftereos 2020 and Horby 2021 provided data on the need for mechanical ventilation and should be studied and included in this review since it is an important outcome."

A: We included these studies in our systematic review. Thanks for your comments.

Reviewer 2 Report

Dear authors,

Your systematic review and meta-analysis evaluated the efficacy of colchicine in the treatment of patients with Covid-19 infection. I found that the article is well structured, deals with an interesting topic and gives readers useful information. However, I have some comments:

As for the introduction:

  1. Does the colchicine have been approved for the treatment of COVID-19 in your country as well as USA, Europe, China or rest of the word? Please provided a more detailed introduction on this point and if regulatory agencies have already taken any decision.
  2. Line 52: through a systematic review “and a meta-analysis”

As for the methods:

  1. PRISMA 2020 statement - Please report in the supplementary material the PRISMA checklist with the respective pages where you reported the information. http://prisma-statement.org/documents/PRISMA_2020_checklist.pdf
  2. Use the word “comparator” instead of “control” when referring to observational studies.
  3. Line 71: What do you mean with “standard care”? What treatments did you included? Please provided in the text or in a separated table.

As for the results:

  1. Line 127: What do you mean with “registers”? Articles?
  2. Line 136: Please provide information about the retrospective or prospective nature of observational studies as well as the median follow up.
  3. Table 1. I suggest you reformat the table
  4. Table 1: Control/Comparator
  5. Table 1: What about other treatments? are these treatment in combination with colchicine or to other treatments assess in the study?
  6. As reported in many articles, male and female patients have a different mortality and a different immune profile that could affect response to treatments (PMID: 34454035). Please report the number of females included for each study as well as median age in Table 1 given that colchicine is an immunomudulatory drug. Moreover, does any of these studies reported any significance difference on outcomes respect to women and men? If not, please discuss briefly in the discussion.

As for the discussion:

  1. You stated that there are different studies (Salah et al, Golpour et al…) reporting different conclusions on mortality about colchicine. Did you included also the studies reported in the previous published review?
  2. Given the wide amount of systematic review and meta-analysis already published, please report the innovative aspects of your work with the respect to the others. It is not clear what this review adds to other works on this topic (i.e. Please discuss more deeply if the other cited reviews have different inclusion criteria with respect to yours)
  3. Line 275: I strongly suggest you report in the results the number of female and age in Table 1 as well as number of those admitted in Intensive care units. Of course, the difference could be related to bias, but I suggest looking also at the different characteristics of patients included in RTC and observational studies that could not the same and could explain what you pointed out in the discussion and conclusion of your manuscript.
  4. Line 279: I think results of NCT should be reported in the result section before discussion. Moreover, why did you limited the search to 31 July 2021 while the search on PubMed and other database arrived at 10 March 2022?

Author Response

Dear Reviewer. 

We answer your comments. 

Introduction

1. Does the colchicine have been approved for the treatment of COVID-19 in your country as well as USA, Europe, China or rest of the word? Please provided a more detailed introduction on this point and if regulatory agencies have already taken any decision.

R: It should be considered that the treatment has not been part of the basic scheme of care for patients with COVID-19 and was only experimental. That is why in many countries its use was not recommended, but it was not restricted either. We added in introduction: “In our country, Peru, treatment with Colchicine has not been established for outpatients or hospitalized patients with COVID-19.”

2. Line 52: through a systematic review “and a meta-analysis”

A: Thanks for your comments. We corrected.

Methods

1. PRISMA 2020 statement - Please report in the supplementary material the PRISMA checklist with the respective pages where you reported the information. http://prisma-statement.org/documents/PRISMA_2020_checklist.pdf

A: Thanks, we corrected the PRISMA Checklist

2. Use the word “comparator” instead of “control” when referring to observational studies.

A: Thanks, we applied this word. 

3. Line 71: What do you mean with “standard care”? What treatments did you included? Please provided in the text or in a separated table.

A: The term "Standard of care" or "usual care" refers to the basic support treatments, and is established independently in each study, or even some do not refer to it, as in our case. 

To avoid complexities, we had to use the term that includes all control or purchaser studies.

As for the results:

1. Line 127: What do you mean with “registers”? Articles?

A: Thanks. We changed the word. 

2. Line 136: Please provide information about the retrospective or prospective nature of observational studies as well as the median follow up.

A: Thanks, we added: “The reported follow-up time between NRSI and RCTs was between 21 and 28 days”.

3. Table 1. I suggest you reformat the table

4. Table 1: Control/Comparator

A: Thanks. We added this term.

5. Table 1: What about other treatments? are these treatment in combination with colchicine or to other treatments assess in the study?

A: These are the additional treatments in either arm reported in the included studies. We investigated between the supplementary material and the original article. 

6. As reported in many articles, male and female patients have a different mortality and a different immune profile that could affect response to treatments (PMID: 34454035). Please report the number of females included for each study as well as median age in Table 1 given that colchicine is an immunomudulatory drug. Moreover, does any of these studies reported any significance difference on outcomes respect to women and men? If not, please discuss briefly in the discussion.

A: Thanks. We added: “It is important to note that, both in our analysis and in the analyses of the studies included in the systematic review, mortality has not been evaluated in a sex-matched method, but was reported in a general way”.

As for the discussion:

1. You stated that there are different studies (Salah et al, Golpour et al…) reporting different conclusions on mortality about colchicine. Did you included also the studies reported in the previous published review?

A: Thanks. This section refers to previous systematic reviews. We refer directly to the main outcome assessed, for any of the designs.

2. Given the wide amount of systematic review and meta-analysis already published, please report the innovative aspects of your work with the respect to the others. It is not clear what this review adds to other works on this topic (i.e. Please discuss more deeply if the other cited reviews have different inclusion criteria with respect to yours)

A: Thanks. Regarding to this point, we added: “However, owing to the differences in technique, the results of numerous systematic reviews differ from ours.”. Then, we began to describe each study and the differences with our work.

3. Line 275: I strongly suggest you report in the results the number of female and age in Table 1 as well as number of those admitted in Intensive care units. Of course, the difference could be related to bias, but I suggest looking also at the different characteristics of patients included in RTC and observational studies that could not the same and could explain what you pointed out in the discussion and conclusion of your manuscript.

A: Thanks. We added the Gender: Male in the table 1. 

4. Line 279: I think results of NCT should be reported in the result section before discussion. Moreover, why did you limited the search to 31 July 2021 while the search on PubMed and other database arrived at 10 March 2022?

A: Thanks. We deleted this. It is wrong. The search was performed in March 2022. 

Reviewer 3 Report

A very interesting systematic review with metanalysis about the use of colchicine in the management of COVID 19, showing that the evidence suggesting the use of this drug to manage this condition is poor and of very low quality.

I enjoyed reading the paper.

Some other pharmacological characteristics of colchicine could be added in the introduction; here is an article you could add : doi: 10.3390/pharmaceutics14020294.

Good Luck

Author Response

Dear reviewer.

Thanks for your comments. We applied the changes.

Round 2

Reviewer 1 Report

Consider adding Forest Plot of need for ventilator in the revised manuscript. 

Author Response

Dear reviewer, 

Thanks for your comments. 

In this systematic review, the outcome "need for ventilator" has no data to be meta-analyzed.

Reviewer 2 Report

Dear authors,

Here I report the previous comments that still require some corrections/implementation:

  1. Does the colchicine have been approved for the treatment of COVID-19 in your country as well as USA, Europe, China or rest of the word? Please provided a more detailed introduction on this point and if regulatory agencies have already taken any decision.

R: It should be considered that the treatment has not been part of the basic scheme of care for patients with COVID-19 and was only experimental. That is why in many countries its use was not recommended, but it was not restricted either. We added in introduction: “In our country, Peru, treatment with Colchicine has not been established for outpatients or hospitalized patients with COVID-19.”

Reviewer: I suggest the authors to move the sentence after reference (6) at page 2. Moreover, try to implement it with proper citations the rational of your study. (DOI: 10.1016/j.rmed.2021.106322)

---------------

  1. Table 1. I suggest you reformat the table

Reviewer: Please try to put the table in a horizontal page

---------------

  1. Line 136: Please provide information about the retrospective or prospective nature of observational studies as well as the median follow up.

A: Thanks, we added: “The reported follow-up time between NRSI and RCTs was between 21 and 28 days”.

Reviewer: Given the different results between observational studies and RCT, in the result section please report the range of median follow both for RCT and observational studies.

------------------

  1. As reported in many articles, male and female patients have a different mortality and a different immune profile that could affect response to treatments (PMID: 34454035). Please report the number of females included for each study as well as median age in Table 1 given that colchicine is an immunomudulatory drug. Moreover, does any of these studies reported any significance difference on outcomes respect to women and men? If not, please discuss briefly in the discussion.

A: Thanks. We added: “It is important to note that, both in our analysis and in the analyses of the studies included in the systematic review, mortality has not been evaluated in a sex-matched method, but was reported in a general way”.

Reviewer: Men with COVID-19 present more severe symptoms, higher mortality and different immune-profile that could affect response to treatments respect to women. However, as reported in a recent review (https://doi.org/10.1016/j.phrs.2021.105848) also in this case,  clinical studies often fail to report results by sex (in COVID-19). I suggest the authors to discuss in the discussion section this important issue also on the basis of different results coming from RCT and observational studies.

------------------

  1. Given the wide amount of systematic review and meta-analysis already published, please report the innovative aspects of your work with the respect to the others. It is not clear what this review adds to other works on this topic (i.e. Please discuss more deeply if the other cited reviews have different inclusion criteria with respect to yours)

A: Thanks. Regarding to this point, we added: “However, owing to the differences in technique, the results of numerous systematic reviews differ from ours.”. Then, we began to describe each study and the differences with our work.

Reviewer: What do you mean by differences in technique? Please be more clear on this point reporting in the description of the previously conducted systematic review and meta-analysis the inclusion criteria and how they differ from yours

-------------------

Author Response

Dear reviewer,

I enclose our response:

Here I report the previous comments that still require some corrections/implementation:

  1. Does the colchicine have been approved for the treatment of COVID-19 in your country as well as USA, Europe, China or rest of the word? Please provided a more detailed introduction on this point and if regulatory agencies have already taken any decision.

R: It should be considered that the treatment has not been part of the basic scheme of care for patients with COVID-19 and was only experimental. That is why in many countries its use was not recommended, but it was not restricted either. We added in introduction: “In our country, Peru, treatment with Colchicine has not been established for outpatients or hospitalized patients with COVID-19.”

Reviewer: I suggest the authors to move the sentence after reference (6) at page 2. Moreover, try to implement it with proper citations the rational of your study. (DOI: 10.1016/j.rmed.2021.106322)

 A: Thanks, we have corrected it. It is possible that it is the trial version that you read, since in the document, the sentence mentioned corresponds to the objectives and is the last paragraph.

---------------

  1. Table 1. I suggest you reformat the table

Reviewer: Please try to put the table in a horizontal page

A: Thanks, Our table 1 is in horizontal version. It is likely that the journal will send in a different format when uploaded as a proof for reviewers.

---------------

  1. Line 136: Please provide information about the retrospective or prospective nature of observational studies as well as the median follow up.

A: Thanks, we added: “The reported follow-up time between NRSI and RCTs was between 21 and 28 days”.

Reviewer: Given the different results between observational studies and RCT, in the result section please report the range of median follow both for RCT and observational studies.

A: Thanks, we applied the comments in the results.

------------------

  1. As reported in many articles, male and female patients have a different mortality and a different immune profile that could affect response to treatments (PMID: 34454035). Please report the number of females included for each study as well as median age in Table 1 given that colchicine is an immunomudulatory drug. Moreover, does any of these studies reported any significance difference on outcomes respect to women and men? If not, please discuss briefly in the discussion.

A: Thanks. We added: “It is important to note that, both in our analysis and in the analyses of the studies included in the systematic review, mortality has not been evaluated in a sex-matched method, but was reported in a general way”.

Reviewer: Men with COVID-19 present more severe symptoms, higher mortality and different immune-profile that could affect response to treatments respect to women. However, as reported in a recent review (https://doi.org/10.1016/j.phrs.2021.105848) also in this case,  clinical studies often fail to report results by sex (in COVID-19). I suggest the authors to discuss in the discussion section this important issue also on the basis of different results coming from RCT and observational studies.

A: Thanks. We added: “In our meta-analysis, differences were found between the incidence of sex-matched outcomes. However, other reports, both systematic reviews and other study designs, have not further analyzed the gender-matched analysis, so they have not noted a great importance in these differences in treatment with Colchicine.”

------------------

  1. Given the wide amount of systematic review and meta-analysis already published, please report the innovative aspects of your work with the respect to the others. It is not clear what this review adds to other works on this topic (i.e. Please discuss more deeply if the other cited reviews have different inclusion criteria with respect to yours)

A: Thanks. Regarding to this point, we added: “However, owing to the differences in technique, the results of numerous systematic reviews differ from ours.”. Then, we began to describe each study and the differences with our work.

Reviewer: What do you mean by differences in technique? Please be more clear on this point reporting in the description of the previously conducted systematic review and meta-analysis the inclusion criteria and how they differ from yours.

A: Thanks. We have pointed out the differences in our analysis technique in each study described (Salah et al, Golpour et al). 
